# Autoregressive Image Generation with Masked Bit Modeling

**Qihang Yu** [1]  **Qihao Liu** [1]  **Ju He** [1]  **Xinyang Zhang** [1]  **Yang Liu** [1]  **Liang-Chieh Chen** [2] [*]  **Xi Chen** [1] [*]

## Abstract

This paper challenges the dominance of continuous pipelines in visual generation. We systematically investigate the performance gap between discrete and continuous methods. Contrary to the belief that discrete tokenizers are intrinsically inferior, we demonstrate that the disparity arises primarily from the total number of bits allocated in the latent space (*i.e.*, the compression ratio). We show that scaling up the codebook size effectively bridges this gap, allowing discrete tokenizers to match or surpass their continuous counterparts. However, existing discrete generation methods struggle to capitalize on this insight, suffering from performance degradation or prohibitive training costs with scaled codebook. To address this, we propose *masked Bit AutoRegressive modeling (BAR)*, a scalable framework that supports arbitrary codebook sizes. By equipping an autoregressive transformer with a *masked bit modeling* head, BAR predicts discrete tokens through progressively generating their constituent bits. BAR achieves a new state-of-the-art gFID of **0.99** on ImageNet-256, outperforming leading methods across both continuous and discrete paradigms, while significantly reducing sampling costs and converging faster than prior continuous approaches. Project page is available at https://bar-gen.github.io/

## 1. Introduction

Visual generative models have driven remarkable progress across a wide range of computer vision tasks (Wang et al., 2023; Team, 2024; Cui et al., 2025; Deng et al., 2025; Wiedemer et al., 2025; Agarwal et al., 2025). A central component of these systems is visual tokenization, which compresses high-dimensional pixel inputs into compact latent repre-

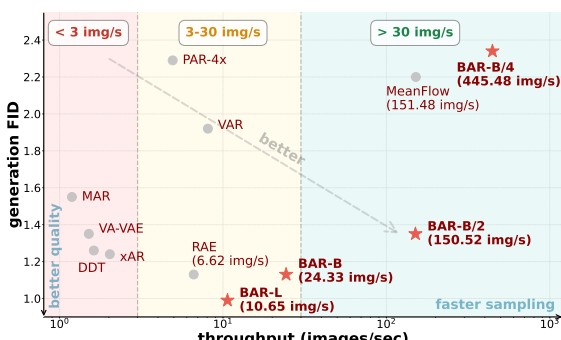

*Figure 1.* **The proposed BAR achieves a superior quality-cost trade-off (generation FID *vs*. throughput) on ImageNet-256.**

sentations. Operating on these latent tokens, a generative model learns the underlying image distribution to synthesize high-fidelity visual content.

Depending on quantization and regularization strategies, visual tokenization and generation pipelines can be broadly categorized into *discrete* and *continuous* approaches. Each paradigm offers distinct advantages: discrete tokenizers align naturally with language modeling, making them suitable for native multimodal large language models (Team, 2024; Cui et al., 2025), whereas continuous tokenizers excel at modeling raw visual signals and preserving fine-grained details. Despite progress in both directions, continuous tokenizers, typically with diffusion models, remain dominant in visual generation (Rombach et al., 2022; Peebles & Xie, 2023; Li et al., 2024; Zheng et al., 2025b). This dominance is largely attributed to their higher information capacity, which enables superior reconstruction fidelity and a higher ceiling for generation (Li et al., 2024; Wang et al., 2025b).

In this work, we investigate the performance gap between discrete and continuous pipelines. Our key observation is that this gap is not intrinsically caused by the nature of the representations, but is instead largely associated with differences in the compression rates used in practice. To make this comparison explicit, we unify both paradigms under a common metric: the number of bits used to represent the latent space. From this unified perspective, we find that the commonly observed inferior performance of discrete tokenizers is largely attributable to their substantially higher compression ratios, which lead to severe information loss. Empirically, we show that allocating more bits per token

---

[1]Amazon FAR (Frontier AI & Robotics) [2]Work done while at FAR. *: Equal advising. Correspondence to: Qihang Yu <yuqiha@amazon.com>.

*Proceedings of the 43rd International Conference on Machine Learning*, Seoul, South Korea. PMLR 306, 2026. Copyright 2026 by the author(s).

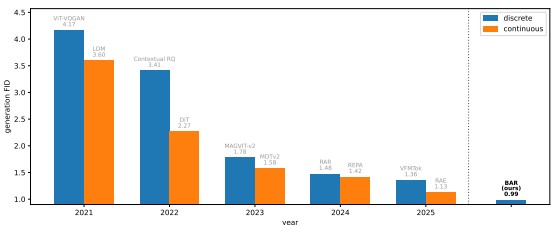

*Figure 2.* **Best discrete and continuous generator comparison.**

(equivalent to scaling up the codebook size) allows discrete tokenizers to match, and in some cases surpass, their continuous counterparts in reconstruction quality.

While increasing the codebook size narrows the reconstruction performance gap, it poses a significant challenge for generative modeling. Discrete generators are typically trained with cross-entropy objectives, and large vocabularies substantially increase both computational and statistical complexity. In particular, scaling the codebook size makes training prohibitively memory-intensive (Han et al., 2025) and increasingly difficult to optimize (Yu et al., 2024a). To address this, we propose replacing the standard linear prediction head with a lightweight ***bit generation mechanism***. Instead of classifying over a massive vocabulary, our method predicts discrete tokens by progressively generating their constituent bits. This design effectively accommodates unbounded vocabulary sizes and consistently improves generation performance, particularly as the codebook size scales.

In summary, discrete tokenizers can serve as competitive visual compressors relative to their continuous counterparts, and that discrete generators can outperform diffusion models in generation fidelity while achieving faster convergence and higher sampling throughput. Building on these findings, we propose *masked **B**it **A**utoreg**R**essive modeling (**BAR**)*, a strong discrete visual generation framework that challenges the prevailing dominance of continuous pipelines. BAR establishes a new state of the art: with only 415M parameters, it achieves a gFID of 1.13 on ImageNet-256 (Deng et al., 2009), surpassing prior discrete models while being $3.68\times$ faster than leading continuous approaches (Zheng et al., 2025b). Additionally, our efficient variant matches the performance of one-step model MeanFlow (Geng et al., 2025) with a $2.94\times$ speedup. Finally, our best-performing variant attains a gFID of 0.99, setting a new benchmark across both discrete and continuous paradigms.

## 2. Related Work

**Continuous Visual Tokenization and Generation.** Continuous visual tokenization and generation pipelines typically consist of two main components: a variational autoencoder (VAE) (Kingma & Welling, 2014) and a diffusion model (Sohl-Dickstein et al., 2015; Song & Ermon, 2019; Ho et al., 2020). VAEs are autoencoders trained with spe-

cific regularization on the latent space (*e.g.*, KL regularization (Rombach et al., 2022)). They usually downsample visual inputs along spatial dimensions while expanding channel dimensions, thereby providing a more compact and structured representation space that is well suited for diffusion-based generation. While a large body of work has focused on diffusion model architectures (Peebles & Xie, 2023; Bao et al., 2023; Gao et al., 2023; Liu et al., 2024; Wang et al., 2025a), denoising trajectories (Lipman et al., 2022; Ma et al., 2024; Liu et al., 2025), and prediction objectives (Li et al., 2024; Ren et al., 2024; 2025; He et al., 2025), SD-VAE (Rombach et al., 2022) has remained the de facto standard VAE backbone in most studies. More recently, increasing attention has been paid to enriching the semantic content of VAE latent spaces, either by incorporating off-the-shelf models (Yao et al., 2025) or by using frozen encoders as tokenizers (Zheng et al., 2025b). There are also works (Hoogeboom et al., 2024; Li et al., 2025; Li & He, 2025) that explore tokenizer-free diffusion models operating in pixel space.

**Discrete Visual Tokenization and Generation.** Building on the foundation of VQGAN (Esser et al., 2021), a substantial body of work has focused on quantizer, the core component of discrete pipelines. One stream of research aims to enhance the utilization and training dynamics of vanilla vector quantization with learnable codebooks (Yu et al., 2022; Zheng & Vedaldi, 2023; Zhu et al., 2024). Conversely, other approaches abandon learnable codebooks entirely in favor of "lookup-free" quantizers (Mentzer et al., 2024; Yu et al., 2024a; Zhao et al., 2025). Notably, while these approaches tokenize images into "bit tokens," they primarily emphasize the benefits of lookup-free quantization, and do not exploit this bit-level structure to redefine the generation targets.

Among these studies, the most closely related works are MaskBit (Weber et al., 2024) and Infinity (Han et al., 2025). MaskBit (Weber et al., 2024) adopts LFQ (Yu et al., 2024a) as the tokenizer and directly feeds bit tokens into the generator. However, it still predicts codebook indices rather than bits during generation, which limits scalability with respect to codebook size, similar to standard discrete generative models. Infinity (Han et al., 2025) scales to extremely large codebook sizes ($2^{64}$) using BSQ (Zhao et al., 2025) and directly generates images from bits. Nevertheless, it relies heavily on the VAR generator (Tian et al., 2024) and an external bit-corrector as a post-processing module. In contrast, the proposed BAR framework is compatible with arbitrary autoregressive formulations and generates bit tokens correctly in a fully self-contained manner, enabled by the proposed masked bit modeling head.

# 3. Method

## 3.1. Background

We begin by introducing the visual tokenization process. A visual tokenizer, whether discrete or continuous, can be viewed as an autoencoder (Hinton & Salakhutdinov, 2006) equipped with an information bottleneck (Kingma & Welling, 2014; Esser et al., 2021; Mentzer et al., 2024). Structurally, it consists of three key components: an encoder $Encoder$, a bottleneck module $Bottleneck$, and a decoder $Decoder$. The nature of the bottleneck distinguishes the two paradigms: a *discrete* bottleneck maps latent features to entries in a finite codebook, whereas a *continuous* bottleneck typically employs dimensionality reduction coupled with regularization, such as the KL-divergence penalty.

Given an input image $\mathbf{I} \in \mathbb{R}^{H \times W \times 3}$, where $H$ and $W$ denote the image height and width, respectively, the encoder first maps the image to a dense feature map $\mathbf{L}$:

$$\mathbf{L} = Encoder(\mathbf{I}), \tag{1}$$

where $\mathbf{L}$ is the encoded feature with spatial shape $\frac{H}{f} \times \frac{W}{f}$.

This feature map is then processed by the bottleneck module $Bottleneck$ to yield the latent representation $\mathbf{X}$. This step imposes paradigm-specific constraints—such as quantization for discrete models or KL-regularization for continuous ones. Finally, the decoder $Decoder$ reconstructs the image $\hat{\mathbf{I}}$ from these latents:

$$\mathbf{X} = Bottleneck(\mathbf{L}), \quad \hat{\mathbf{I}} = Decoder(\mathbf{X}). \tag{2}$$

In practice, each latent token $x \in \mathbf{X}$ is encouraged to follow a structured distribution (*e.g.*, discrete or Gaussian), which facilitates subsequent generative modeling by making the latent space easier to model and sample from.

## 3.2. Benchmarking Discrete and Continuous Tokenizers

The primary distinction between discrete and continuous tokenizers lies in the design of the bottleneck. Discrete tokenizers typically rely on codebook lookup with hard assignments to discretize latent features, whereas continuous tokenizers impose bottlenecks through dimensionality reduction combined with regularization losses. This fundamental difference makes direct comparison between the two paradigms nontrivial. In practice, discrete tokenizers are commonly characterized by their codebook size (Esser et al., 2021; Yu et al., 2022), while continuous tokenizers are often compared based on the dimensionality of their latent representations (Rombach et al., 2022; Li et al., 2024).

To enable a unified and fair comparison across these paradigms, we evaluate both tokenizers using a common metric: the *Bit Budget* ($B$). This metric quantifies the total information capacity allocated to the latent space, serving as

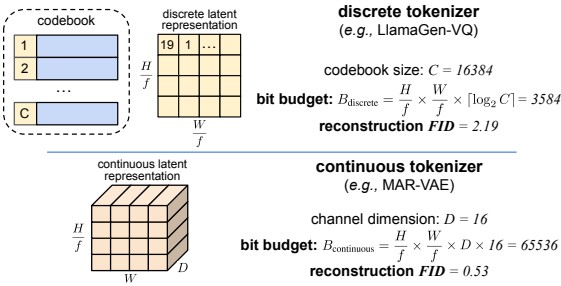

*Figure 3.* **A unified view for comparing discrete and continuous tokenizers.** By measuring information capacity in bits, we enable a direct comparison. The continuous tokenizer MAR-VAE (Li et al., 2024) outperforms the discrete tokenizer LlamaGen-VQ (Sun et al., 2024) in reconstruction quality, a result directly attributable to its substantially higher bit allocation.

a proxy for the nominal compression ratio. Formally, consider an input image $\mathbf{I}$ of height $H$ and width $W$, processed by a tokenizer with spatial downsampling factor $f$. For a discrete tokenizer with codebook size $C$, its bit budget is[1]:

$$B_{\text{discrete}} = \frac{H}{f} \times \frac{W}{f} \times \lceil \log_2 C \rceil. \tag{3}$$

Conversely, for a continuous tokenizer with latent channel dimension $D$, the bit budget is:

$$B_{\text{continuous}} = \frac{H}{f} \times \frac{W}{f} \times 16D, \tag{4}$$

where the constant factor 16 reflects mixed-precision training, with each latent channel represented using 16 bits. While bit budget $B$ defines the nominal capacity, the effective information content may be lower due to dead codebook entries or distributional regularization. This metric facilitates the direct comparison shown in Fig. 3.

## 3.3. Discrete Tokenizers Beat Continuous Tokenizers

Equipped with the Bit Budget metric, we conduct a systematic evaluation of existing discrete and continuous tokenizers. As shown in Fig. 4, we observe a distinct separation between the two paradigms: discrete tokenizers generally exhibit worse reconstruction quality while using substantially fewer bits. This discrepancy in compression ratio is non-negligible and can largely account for the inferior reconstruction performance observed in discrete methods.

Crucially, we identify a convergence trend: as we increase the number of bits allocated to the latent space, the performance of discrete tokenizers progressively improves, narrowing the gap with continuous tokenizers. This observation prompts a critical investigation: *Is the perceived inferiority*

---

[1]We mainly discuss the most common single-scale and single-codebook tokenizers, whereas the formulation can be easily generalized to other cases such as multi-scale (Tian et al., 2024) or multi-codebook (Qu et al., 2025; Ma et al., 2025a).

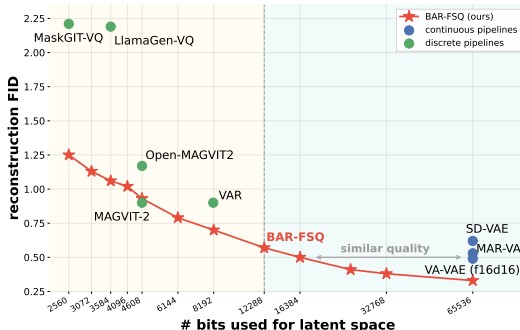

*Figure 4.* **Scaling BAR's discrete tokenizer (BAR-FSQ) with Bit Budget.** Standard discrete methods (green circles) historically lag behind continuous baselines (blue circles) primarily due to restricted bit allocation. By systematically scaling the codebook size, BAR-FSQ (red curve) demonstrates that discrete tokenizer's reconstruction performance is not inherently bounded; it matches and further surpasses continuous reconstruction fidelity with increased bit budget, challenging the assumption that continuous latent spaces are required for high-fidelity reconstruction.

*of discrete tokenizers intrinsic to the quantization bottleneck, or is it merely a consequence of insufficient bit allocation?*

To address this, we examine the effect of scaling the codebook size to approach the bit budget of continuous tokenizers. Since classical Vector Quantization (VQ) with learnable codebooks becomes computationally infeasible at extreme scales (*e.g.*, $2^{256}$), we adopt the FSQ quantizer (Mentzer et al., 2024)[2]. This allows us to scale smoothly without auxiliary quantization or entropy losses (Chang et al., 2022; Yu et al., 2024a; Zhao et al., 2025).

For simplicity, we fix the number of latent tokens to 256 (*i.e.*, a downsampling ratio of $f = 16$ for $256 \times 256$ images) and use 1 bit per channel in the FSQ quantizer. We then vary the latent channel dimension from 10 to 12, 14, 16, 18, 32, 64, 128, and 256, corresponding to codebook sizes of $2^{10}$, $2^{12}$, $2^{14}$, $2^{16}$, $2^{18}$, $2^{32}$, $2^{64}$, $2^{128}$, and $2^{256}$, respectively.

The results, shown by the BAR-FSQ curve in Fig. 4, demonstrate that reconstruction quality improves consistently with codebook size. Notably, when the bit budget increases beyond certain point, the discrete tokenizer achieves competitive or superior fidelity. For instance, at a budget of 65536 bits, our discrete tokenizer attains an rFID of 0.33, outperforming the SD-VAE (rFID 0.62).

Furthermore, discrete tokenizers demonstrate superior efficiency in budget utilization. With only 16384 bits, we achieve comparable performance (rFID 0.50). This indicates that discrete method yields highly expressive representations even under strict constraints, leading to our core discovery:

---

[2]The discussion here can easily generalize to other bit quantization such as LFQ (Yu et al., 2024a) or BSQ (Zhao et al., 2025).

*The main performance bottleneck of discrete tokenizer lies in an insufficient bit budget, while scaling up codebook size enables discrete tokenization outperform continuous approaches.*

### 3.4. Discrete Autoregressive Models Beat Diffusion

While scaling the codebook size effectively resolves the reconstruction bottleneck (as established in the preceding subsection), it introduces a new, critical impediment to generative modeling: the *vocabulary scaling problem*.

Standard autoregressive models face a prohibitive computational cliff as vocabularies expand. Projecting high-dimensional hidden states onto a vocabulary of millions ($2^{20}$) or billions ($2^{30}$) of entries renders the final linear prediction head intractable in terms of both memory and compute. Consequently, prior works typically cap codebook sizes at $2^{18}$ (262144), accepting a ceiling on reconstruction fidelity to preserve trainability. Furthermore, even when hardware permits, learning a reliable categorical distribution over such a vast space is statistically difficult, leading to a sharp degradation in generation quality (Yu et al., 2024a).

We empirically validate this limitation by training models with a standard *linear* prediction head across different codebook sizes. The model works fine with limited codebook size but stops at 18 bits (corresponding to vocabulary sizes 262144); beyond this range, training becomes unaffordable under typical GPU memory constraints. We also experimented with a *bits-based* head that predicts the bit representation of target discrete token instead of the index over entire vocabulary (Han et al., 2025). While this approach enables training with large codebook sizes, it consistently yields inferior performance across vocabulary sizes and suffers from severe degradation as the vocabulary scales.

**Prediction Head as a Bit Generator.** To overcome this, we disentangle the generator into two distinct functional components: an *Autoregressive Transformer*, which captures global structure via causal attention, and a *Prediction Head*, which projects latent embeddings onto specific discrete codes. This separation is critical: as codebook sizes scale, the autoregressive transformer remains computationally invariant; the entire burden of the exponential vocabulary growth is absorbed exclusively by the prediction head.

Unlike prior approaches relying on *linear* or *bit-based* projection, we propose a paradigm shift: rather than treating token prediction as a massive classification task, we formulate it as a conditional generation task. We introduce a **Masked Bit Modeling (MBM) Head**, which generates the target discrete token via an iterative, bit-wise unmasking process conditioned on the *autoregressive transformer*'s output. The proposed prediction head is lightweight, typically requiring only a small number of additional forward passes

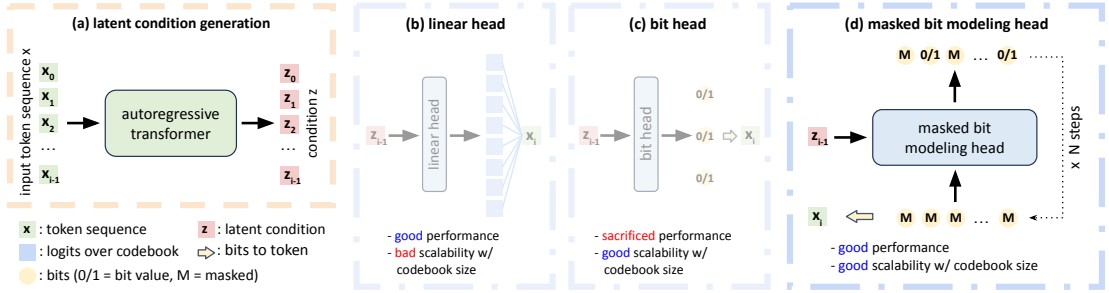

*Figure 5.* **Overview of the proposed BAR framework.** We decompose autoregressive visual generation into two stages: context modeling and token prediction. (a) For context modeling, we employ an autoregressive transformer to generate latent conditions via causal attention. For the subsequent token prediction stage, we contrast our method with two baselines: (b) A standard linear head predicts logits over the full codebook. While effective for small vocabularies ($< 2^{18}$), it fails to scale to larger sizes due to computational bottlenecks. (c) A bit-based head predicts bits directly; while scalable, it results in inferior generation quality. (d) The proposed Masked Bit Modeling (MBM) head generates bits via a progressive unmasking mechanism conditioned on the autoregressive transformer's output. Unlike the baselines, MBM achieves both exceptional scalability and superior generation quality.

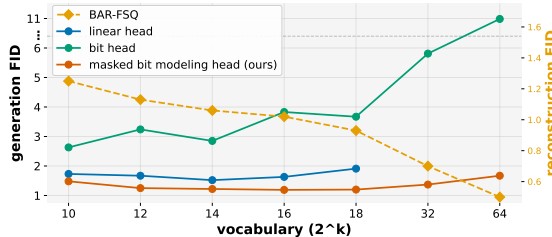

*Figure 6.* **Reconstruction and generation quality as a function of BAR tokenizer's vocabulary size.** Unlike the linear head, the proposed *masked bit modeling head* scales to arbitrary codebook sizes. Furthermore, it achieves a superior reconstruction–generation trade-off compared to the bit head.

to decode a discrete token.

**Formulation.** Let $\mathcal{F}$ denote the autoregressive transformer (Vaswani et al., 2017). Given a causal prefix of discrete tokens $\{x_1, x_2, \ldots, x_{i-1}\}$, where each token $x$ is represented by $k$-bit binary code, the autoregressive transformer maps the input to a sequence $\{z_1, z_2, \ldots, z_{i-1}\}$. Specifically, for the prediction of the $i$-th token, we have:

$$z_{i-1} = \mathcal{F}(\{x_1, x_2, \ldots, x_{i-1}\}), \quad (5)$$

We utilize $z_{i-1}$ as a condition to predict the next token $x_i$ via a masked bit modeling head $\mathcal{G}$ parameterized by $\theta$:

$$\hat{x}_i = \mathcal{G}_\theta\big(\text{Mask}_{bit}(x_i) \mid z_{i-1}, \mathcal{M}\big), \quad (6)$$

where $\text{Mask}_{bit}(\cdot)$ randomly masks a subset of bits in $x_i$ by replacing them with a special mask token, and $\mathcal{M}$ denotes the masking ratio.

During training, we optimize cross-entropy loss between the predicted token $\hat{x}_i$ and the ground-truth token $x_i$:

$$\mathcal{L} = \frac{1}{n} \sum_{i=1}^{n} \text{CrossEntropy}_{bit}(x_i, \hat{x}_i), \quad (7)$$

where $n$ is the sequence length, and $\text{CrossEntropy}_{bit}$ applies the loss in a bit-wise manner.

At inference, the next token is not selected via a single sampling step but is "generated" through a progressive bit-wise unmasking schedule (Chang et al., 2022).

As illustrated in Fig. 5, this design offers two key advantages. First, in terms of **scalability**, decomposing the token into its constituent bits bypasses the need for a monolithic softmax over the entire vocabulary, reducing memory complexity from $\mathcal{O}(C)$ to $\mathcal{O}(\log_2 C)$, where $C = 2^k$ is the codebook size. Second, regarding **robustness**, the bit-wise masking acts as a strong regularizer that consistently improves generation quality. As a result, the MBM head yields a superior trade-off between reconstruction (rFID) and generation (gFID) across all codebook scales, as shown in Fig. 6.

**Discussion.** Unlike standard linear heads, which are constrained by fixed computational costs and memory requirements that scale linearly with vocabulary size, the proposed masked bit modeling head facilitates discrete generation with arbitrarily large vocabularies. Our results demonstrate consistent improvements over baselines, with the advantage becoming more pronounced at larger codebook sizes. This improved scaling behavior stems from the model's capacity to flexibly allocate more computation per token via progressive unmasking, a mechanism analogous to the iterative denoising process in diffusion models.

## 4. Experimental Results

### 4.1. Implementation Details

**Tokenizer.** We build the discrete tokenizer using FSQ (Mentzer et al., 2024), incorporating several modern design choices from recent works (Weber et al., 2024; Tian et al., 2024; Lu et al., 2025; Zheng et al., 2025b) to better align with contemporary training recipes. Specifically,

*Table 1.* **Scaling BAR-FSQ codebook size ($C$) with different prediction heads.** Unlike linear and bit-based baselines, the proposed *masked bit modeling* (MBM) scales to arbitrary codebook sizes while delivering superior generation quality.

| bits | $C$ | rFID | prediction head | w/o CFG | | w/ CFG | |
|---|---|---|---|---|---|---|---|
| | | | | gFID↓ | IS↑ | gFID↓ | IS↑ |
| 10 | 1024 | 1.25 | linear | **2.80** | 171.6 | 1.73 | 238.7 |
| | | | bit | 10.77 | 107.6 | 2.63 | 213.9 |
| | | | **MBM** | 3.10 | **180.6** | **1.48** | **271.2** |
| 12 | 4096 | 1.13 | linear | 2.70 | 180.3 | 1.67 | 248.9 |
| | | | bit | 14.52 | 100.1 | 3.24 | 213.9 |
| | | | **MBM** | **2.10** | **207.9** | **1.25** | **268.3** |
| 14 | 16384 | 1.06 | linear | 2.60 | 192.4 | 1.52 | 263.0 |
| | | | bit | 14.57 | 96.9 | 2.85 | 224.5 |
| | | | **MBM** | **1.71** | **240.8** | **1.22** | **292.2** |
| 16 | 65536 | 1.02 | linear | 2.90 | 182.7 | 1.63 | 253.2 |
| | | | bit | 23.11 | 71.8 | 3.83 | 215.0 |
| | | | **MBM** | **1.68** | **231.6** | **1.19** | **282.3** |
| 18 | 262144 | 0.93 | linear | 3.45 | 172.9 | 1.91 | 241.0 |
| | | | bit | 21.81 | 78.7 | 3.67 | 223.0 |
| | | | **MBM** | **1.77** | **228.5** | **1.20** | **281.1** |
| 32 | $\sim 4.29 \times 10^9$ | 0.70 | linear | OOM | OOM | OOM | OOM |
| | | | bit | 45.11 | 49.6 | 5.81 | 204.3 |
| | | | **MBM** | **2.37** | **197.8** | **1.37** | **292.1** |
| 64 | $\sim 1.84 \times 10^{19}$ | 0.50 | linear | OOM | OOM | OOM | OOM |
| | | | bit | 73.67 | 30.7 | 10.97 | 183.0 |
| | | | **MBM** | **2.60** | **213.6** | **1.67** | **295.0** |

we initialize the encoder from SigLIP2-so400M (Tschannen et al., 2025) and apply an L2 loss against the original CLIP features to encourage semantic alignment in the latent space (Zheng et al., 2025a). For the decoder, we use a ViT-L (Dosovitskiy et al., 2021) model trained from scratch, and we employ a frozen DINO model (Caron et al., 2021; Sauer et al., 2023; Tian et al., 2024; Zheng et al., 2025b) as the discriminator. The final training objective combines L1, L2, perceptual (Zhang et al., 2018), Gram loss (Lu et al., 2025) and GAN losses (Goodfellow et al., 2014). The training is conducted for 40 epochs for ablation studies. For final models, we finetune the decoder for 40 more epochs.

**Generator.** We build upon the state-of-the-art discrete autoregressive generation model RAR (Yu et al., 2025a). In addition, we augment the model with several architectural components commonly used in recent diffusion-based generators (Yao et al., 2025; Li & He, 2025; Zheng et al., 2025b), including RoPE (Su et al., 2024), SwiGLU (Shazeer, 2020), RMSNorm (Zhang & Sennrich, 2019), and repeated class conditioning (Li & He, 2025). The masked bit modeling head employs a 3-layer SwiGLU with adaLN (Ba et al., 2016; Peebles & Xie, 2023), which is lightweight and incurs only marginal extra cost. All training hyperparameters strictly follow the original RAR configuration. Training is conducted for 400 epochs with a batch size of 2048.

**Sampling.** We sample 50000 images for FID computation using the evaluation code from (Dhariwal & Nichol, 2021). When classifier-free guidance is employed, we adopt a simple linear guidance schedule (Chang et al., 2023).

### 4.2. Ablation Studies

We study the impact of different designs based on BAR-B, supported by results both without and with classifier-free guidance (CFG) (Ho & Salimans, 2022) for a comprehensive analysis of how different designs affect performance.

**Different Prediction Heads.** As shown in Tab. 1, the linear head performs reasonably well when the codebook size is small, but it does not scale to large codebook sizes: when the vocabulary reaches $2^{32}$, training is no longer feasible within a reasonable resource budget. Although the bits head (Han et al., 2025) partially alleviates the computational bottleneck by making generation with large vocabularies affordable, its generation quality is significantly inferior. Without CFG, all bits-head variants yield gFID values $> 10$, and even with CFG, performance remains poor with gFID $> 2.6$, indicating substantial degradation in generation quality. Besides, its performance degrades as vocabulary scales.

In contrast, the proposed masked bit modeling head not only scales naturally to arbitrary codebook sizes, but also consistently yields superior generation performance. Even with a large codebook of size $2^{32}$, it achieves a gFID of 1.37, approaching state-of-the-art performance.

**Masking Ratio Sampling Strategy.** We evaluate different masking ratio sampling strategies during training in Tab. 2a, specifically comparing arccos (Besnier & Chen, 2023), uniform, and logit-normal sampling (Esser et al., 2024). In contrast to typical Masked Image Modeling (MIM) generative models (Chang et al., 2022; Besnier & Chen, 2023; Yu et al., 2024b; Weber et al., 2024), which often favor tail-heavy distributions (*e.g.*, arccos), BAR does not require such skewing. Instead, simple uniform sampling performs remarkably well. Overall, BAR demonstrates robustness across all strategies, achieving competitive generation quality in each case. We adopt logit-normal sampling as the default, as it yields a slight performance advantage, particularly for larger codebook sizes.

**Prediction Head Size.** We summarize the impact of prediction head capacity in Tab. 2b, varying both the number of layers and the hidden width. We observe consistent improvements in generation quality without CFG as the prediction head capacity increases, while the gains become less pronounced when CFG is applied. Interestingly, the benefits of a larger prediction head are more substantial for larger codebook sizes, suggesting that predicting discrete tokens from a larger space is inherently more challenging and therefore benefits from a stronger generative prediction head.

**Sampling Strategy.** We ablate sampling strategies in Tab. 2c along two dimensions: the number of sampling steps and the bit unmasking schedule. Increasing the number of sampling steps from 2 to 3 yields a significant improvement in generation quality, while further increasing the steps to 5 or 6 provides only marginal gains. We also evaluate a back-loading bit unmasking schedule and find that it improves performance with CFG, but slightly degrades performance without CFG, where a uniform unmasking schedule remains preferable.

**Efficient Generation with Token-Shuffling.** The proposed MBM head offers an additional advantage by enabling effi-

*Table 2.* **Ablation studies on BAR design.** The rows labeled with gray color indicate our choices for final models.

**(a)** **Impact of masking strategy during training.** BAR demonstrates robustness across different masking strategies.

| bits | masking strategy | w/o CFG | | w/ CFG | |
|---|---|---|---|---|---|
| | | FID↓ | IS↑ | FID↓ | IS↑ |
| 10 | arccos | 4.02 | 162.6 | 1.88 | 229.3 |
| | uniform | 3.04 | 183.4 | 1.47 | 275.8 |
| | logit-normal | 3.10 | 180.6 | 1.48 | 271.2 |
| 12 | arccos | 2.33 | 192.5 | 1.38 | 295.9 |
| | uniform | 2.11 | 206.4 | 1.27 | 268.7 |
| | logit-normal | 2.10 | 207.9 | 1.25 | 268.3 |
| 16 | arccos | 1.89 | 218.1 | 1.46 | 311.8 |
| | uniform | 1.78 | 225.4 | 1.22 | 293.7 |
| | logit-normal | 1.68 | 231.6 | 1.19 | 282.3 |
| 32 | arccos | 2.66 | 192.0 | 1.48 | 287.8 |
| | uniform | 2.48 | 209.0 | 1.38 | 281.9 |
| | logit-normal | 2.37 | 197.8 | 1.37 | 292.1 |

**(b)** **Scaling codebook size with different head sizes.** Increasing the head size improves performance, particularly for larger vocabularies. However, these benefits offer diminishing returns when Classifier-Free Guidance (CFG) is applied.

| bits | head size (#layers × width) | w/o CFG | | w/ CFG | |
|---|---|---|---|---|---|
| | | FID↓ | IS↑ | FID↓ | IS↑ |
| 10 | 3 × 1024 | 3.29 | 176.0 | 1.52 | 284.0 |
| | 3 × 1536 | 3.14 | 179.9 | 1.51 | 260.7 |
| | 3 × 2048 | 3.10 | 180.6 | 1.48 | 257.9 |
| | 6 × 2048 | 2.87 | 187.8 | 1.48 | 265.4 |
| 12 | 3 × 1024 | 2.16 | 206.2 | 1.26 | 286.3 |
| | 3 × 1536 | 2.07 | 195.3 | 1.25 | 280.1 |
| | 3 × 2048 | 2.10 | 207.9 | 1.25 | 268.3 |
| | 6 × 2048 | 2.02 | 198.2 | 1.27 | 279.5 |
| 16 | 3 × 1024 | 1.85 | 216.3 | 1.21 | 304.0 |
| | 3 × 1536 | 1.79 | 222.5 | 1.20 | 291.2 |
| | 3 × 2048 | 1.68 | 231.6 | 1.19 | 282.3 |
| | 6 × 2048 | 1.63 | 233.8 | 1.19 | 293.2 |
| 32 | 3 × 1024 | 2.68 | 198.7 | 1.45 | 282.9 |
| | 3 × 1536 | 2.44 | 205.8 | 1.41 | 278.6 |
| | 3 × 2048 | 2.37 | 197.8 | 1.37 | 292.1 |
| | 6 × 2048 | 2.10 | 210.0 | 1.31 | 290.1 |

**(c)** **Impact of sampling strategy.** More steps advances results, while back-loading schedule further improves with CFG.

| bits | bits unmasking schedule | w/o CFG | | w/ CFG | |
|---|---|---|---|---|---|
| | | FID↓ | IS↑ | FID↓ | IS↑ |
| 16 | [8, 8] | 3.73 | 191.1 | 1.55 | 272.5 |
| | [5, 5, 6] | 1.95 | 218.2 | 1.20 | 291.3 |
| | [4, 4, 4, 4] | 1.68 | 231.6 | 1.19 | 282.3 |
| | [3, 3, 3, 4] | 1.64 | 235.8 | 1.23 | 301.1 |
| | [2, 2, 3, 3, 3, 3] | 1.64 | 230.0 | 1.22 | 307.2 |
| | [2, 2, 5, 7] | 1.81 | 214.2 | 1.15 | 289.2 |

**(d)** **Efficient BAR.** Sampling are based on uniform schedules with 4 bits unmasking steps per token for all methods. BAR enables better accuracy-cost trade-off.

| patch size | token | bits per token | w/o CFG | | w/ CFG | | |
|---|---|---|---|---|---|---|---|
| | | | FID↓ | IS↑ | FID↓ | IS↑ | images / sec |
| BAR-B | 256 | 16 | 1.68 | 231.6 | 1.19 | 282.3 | 24.9 |
| BAR-B/2 | 64 | 64 | 2.24 | 217.3 | 1.35 | 293.4 | 150.3 |
| BAR-B/4 | 16 | 256 | 3.50 | 212.4 | 2.34 | 274.7 | **445.5** |

cient visual generation trading off sequence length and bits per token using patch size, similar to prior practices (Rombach et al., 2022; Peebles & Xie, 2023; Ma et al., 2025b). By shuffling from tokens to bits (for example, flattening and concatenating the bits of neighboring tokens), the effective token sequence length can be significantly reduced, enabling more efficient generation. As shown in Tab. 2d, BAR provides a flexible mechanism to trade off generation quality and computational cost by balancing computation between the autoregressive transformer and the masked bit modeling head. Specifically, BAR-B can downsample the latent space by 2× (named BAR-B/2 with patch size 2), resulting in 4× fewer tokens, while incurring only a modest degradation in performance (from 1.68 to 2.24 without CFG and from 1.19 to 1.35 with CFG). Consequently, sampling throughput increases substantially, from 24.9 images per second to 150.3 images per second. More aggressive downsampling leads to BAR-B/4 (patch size 4), which further improves the sampling speed to 445.5 images per second.

### 4.3. Main Results

We report BAR results against state-of-the-art methods on the ImageNet-1K benchmarks at resolutions 256 × 256. For all results reported, we use the official ADM scripts (Dhariwal & Nichol, 2021) to ensure a fair comparison.

**ImageNet 256×256.** We summarize the results in Tab. 3. We observe that BAR-B, despite having only 415M param-

eters, achieves substantially better performance than prior state-of-the-art discrete generation methods. In particular, BAR-B uses only one quarter of the model size of RAR (415M *vs.* 1.5B), yet attains significantly higher generation quality (gFID 1.13 *vs.* 1.48). It also outperforms other discrete approaches by a clear margin, including VAR (1.13 *vs.* 1.92) and LlamaGen (1.13 *vs.* 2.18).

Moreover, BAR-B already surpasses state-of-the-art diffusion models based on continuous pipelines. Specifically, BAR-B outperforms xAR, which is approximately 3× larger in model size, achieving a gFID of 1.13 compared to 1.24. Despite its compact size, BAR-B exceeds the performance of several strong diffusion-based models, including DDT (1.13 *vs.* 1.26), VA-VAE (1.13 *vs.* 1.35), and MAR (1.13 *vs.* 1.55). Compared to the concurrent work RAE, the two methods achieve comparable performance at gFID 1.13.

Scaling BAR-B to a larger model yields BAR-L, which further improves performance and significantly outperforms all prior methods, both discrete and continuous, achieving a new state-of-the-art gFID of 0.99. Notably, BAR-L not only sets a new record under classifier-free guidance (0.99 *vs.* 1.13 for RAE), but also establishes a new best result without guidance (1.42 *vs.* 1.51 for RAE).

**Sampling Speed.** We compare BAR with state-of-the-art methods in terms of sampling speed in Tab. 4. Notably, the **efficient** variants of BAR achieve an excellent trade-off between generation quality and sampling efficiency. BAR-B/2,

*Table 3.* **ImageNet-1K** $256 \times 256$ **generation results.** We report metrics with and without classifier-free guidance. BAR only adopts a simple linear guidance schedule, with no need for auto-guidance (Karras et al., 2024) from an external model that is used by other state-of-the-art methods (Zheng et al., 2025b).

| method | epochs | #params | generation@256 w/o guidance | | | | generation@256 w/ guidance | | | |
|---|---|---|---|---|---|---|---|---|---|---|
| | | | FID↓ | IS↑ | Prec.↑ | Rec.↑ | FID↓ | IS↑ | Prec.↑ | Rec.↑ |
| *pixel space* | | | | | | | | | | |
| ADM (Dhariwal & Nichol, 2021) | 350 | 554M | 10.94 | 101.0 | 0.69 | 0.63 | 3.94 | 215.8 | 0.83 | 0.53 |
| JiT (Li & He, 2025) | 600 | 2B | - | - | - | - | 1.82 | 292.6 | 0.79 | 0.62 |
| SiD2 (Hoogeboom et al., 2024) | 1280 | - | - | - | - | - | 1.38 | - | - | - |
| *continuous tokens* | | | | | | | | | | |
| DiT (Peebles & Xie, 2023) | 1400 | 675M | 9.62 | 121.5 | 0.67 | 0.67 | 2.27 | 278.2 | 0.83 | 0.57 |
| SiT (Ma et al., 2024) | 1400 | 675M | 8.61 | 131.7 | 0.68 | 0.67 | 2.06 | 270.3 | 0.82 | 0.59 |
| DiMR (Liu et al., 2024) | 800 | 1.1B | 3.56 | - | - | - | 1.63 | 292.5 | 0.79 | 0.63 |
| FlowAR (Ren et al., 2024) | 400 | 1.9B | - | - | - | - | 1.65 | 296.5 | 0.83 | 0.60 |
| MDTv2 (Gao et al., 2023) | 1080 | 676M | - | - | - | - | 1.58 | 314.7 | 0.79 | 0.65 |
| MAR (Li et al., 2024) | 800 | 943M | 2.35 | 227.8 | 0.79 | 0.62 | 1.55 | 303.7 | 0.81 | 0.62 |
| VA-VAE (Yao et al., 2025) | 80 | 675M | 4.29 | - | - | - | - | - | - | - |
| | 800 | | 2.17 | 205.6 | 0.77 | 0.65 | 1.35 | 295.3 | 0.79 | 0.65 |
| REPA (Yu et al., 2025b) | 80 | 675M | 7.90 | 122.6 | 0.70 | 0.65 | - | - | - | - |
| | 800 | | 5.78 | 158.3 | 0.70 | 0.68 | 1.29 | 306.3 | 0.79 | 0.64 |
| DDT (Wang et al., 2025a) | 80 | 675M | 6.62 | 135.2 | 0.69 | 0.67 | 1.52 | 263.7 | 0.78 | 0.63 |
| | 400 | | 6.27 | 154.7 | 0.68 | 0.69 | 1.26 | 310.6 | 0.79 | 0.65 |
| xAR (Ren et al., 2025) | 800 | 1.1B | - | - | - | - | 1.24 | 301.6 | 0.83 | 0.64 |
| RAE (Zheng et al., 2025b) | 80 | 839M | 2.16 | 214.8 | 0.82 | 0.59 | - | - | - | - |
| | 800 | | 1.51 | 242.9 | 0.79 | 0.63 | 1.13 | 262.6 | 0.78 | 0.67 |
| *discrete tokens* | | | | | | | | | | |
| MaskGIT (Chang et al., 2022) | 300 | 177M | 6.18 | 182.1 | - | - | - | - | - | - |
| Open-MAGVIT2 (Luo et al., 2024) | 350 | 1.5B | - | - | - | - | 2.33 | 271.8 | 0.84 | 0.54 |
| LlamaGen (Sun et al., 2024) | 300 | 3.1B | 9.38 | 112.9 | 0.69 | 0.67 | 2.18 | 263.3 | 0.81 | 0.58 |
| TiTok (Yu et al., 2024b) | 800 | 287M | 4.44 | 168.2 | - | - | 1.97 | 281.8 | - | - |
| VAR (Tian et al., 2024) | 350 | 2.0B | - | - | - | - | 1.92 | 323.1 | 0.82 | 0.59 |
| MAGVIT-v2 (Yu et al., 2024a) | 1080 | 307M | 3.65 | 200.5 | - | - | 1.78 | 319.4 | - | - |
| MaskBit (Weber et al., 2024) | 1080 | 305M | - | - | - | - | 1.52 | 328.6 | - | - |
| RAR (Yu et al., 2025a) | 400 | 1.5B | - | - | - | - | 1.48 | 326.0 | 0.80 | 0.63 |
| BAR-B (ours) | 400 | 415M | 1.64 | 230.4 | 0.80 | 0.62 | 1.13 | 289.0 | 0.77 | 0.66 |
| BAR-L (ours) | 80 | 1.1B | **1.71** | 224.3 | 0.80 | 0.63 | **1.15** | 288.7 | 0.77 | 0.66 |
| | 400 | | **1.42** | 236.2 | 0.79 | 0.65 | **0.99** | 296.9 | 0.77 | 0.69 |

*Table 4.* **Sampling throughput (including de-tokenization process).** All are benchmarked using a single H200, with float32 precision. BAR only uses KV-cache without further optimization.

| method | #params | FID↓ | images / sec |
|---|---|---|---|
| PAR-4× (Wang et al., 2025c) | 3.1B | 2.29 | 4.92 |
| VAR (Tian et al., 2024) | 2.0B | 1.92 | 8.08 |
| MeanFlow (Geng et al., 2025) | 676M | 2.20 | 151.48 |
| BAR-B/4 (ours) | 416M | 2.34 | **445.48** |
| BAR-B/2 (ours) | 415M | 1.35 | 150.52 |
| MAR (Li et al., 2024) | 943M | 1.55 | 1.19 |
| VA-VAE (Yao et al., 2025) | 675M | 1.35 | 1.51 |
| DDT (Wang et al., 2025a) | 675M | 1.26 | 1.62 |
| xAR (Ren et al., 2025) | 1.1B | 1.24 | 2.03 |
| RAE (Zheng et al., 2025b) | 839M | 1.13 | 6.62 |
| BAR-B (ours) | 415M | 1.13 | 24.33 |
| BAR-L (ours) | 1.1B | **0.99** | 10.65 |

with a gFID of 1.35, not only outperforms all efficient generation methods such as PAR (gFID 2.29) and VAR (gFID 1.92), but also achieves substantially faster sampling speeds, with $30.59\times$ and $18.64\times$ speedups over PAR and VAR, respectively. Even when compared to single-step diffusion models such as MeanFlow, BAR-B/2 demonstrates superior generation quality (gFID 1.35 *vs.* 2.20) at comparable sampling speed (150.52 *vs.* 151.48 images per second). The most efficient variant, BAR-B/4, achieves generation quality comparable to MeanFlow (gFID 2.34 *vs.* 2.20), while producing samples $2.94\times$ faster.

In more performance-oriented comparisons, BAR-B achieves state-of-the-art visual quality while being $20.45\times$, $16.11\times$, $15.02\times$, $11.99\times$, and $3.68\times$ faster than MAR, VA-VAE, DDT, xAR, and RAE, respectively. Notably, the best-performing variant BAR-L not only sets a new state-of-the-art record with gFID 0.99, but also maintains a clear advantage in sampling speed, achieving $8.95\times$ speedup over MAR, $5.25\times$ over xAR, and $1.61\times$ over RAE.

## 5. Conclusion

In this paper, we presented a unified and fair comparison between discrete and continuous visual tokenizers. We showed that differences in compression ratio, as measured by the number of bits allocated to the latent space, constitute a dominant factor underlying the observed performance differences between discrete and continuous tokenizers. When operating under comparable bit budgets, discrete tokenizers can match or even outperform their continuous counterparts.

Building on this analysis, we introduced a novel generative prediction head that models discrete tokens by generating their bit representations. This design enables efficient and effective discrete generation with arbitrarily large vocabularies, overcoming a key limitation of prior discrete generative models. As a result, the proposed *masked bit autoregressive modeling* framework establishes a new state of the art, sub-

stantially outperforming both existing discrete methods and
strong continuous baselines.

## Impact Statement

This work advances the field of visual generation by demonstrating that discrete tokenizers can match or surpass continuous approaches when given sufficient information capacity, while achieving faster sampling speeds and more efficient training. By making high-quality image generation more computationally accessible through such methods, this research could democratize generative AI for researchers with limited resources and reduce the environmental impact of large-scale generation tasks. However, the improved quality and efficiency of these models also amplify concerns around potential misuse for creating deepfakes, spreading misinformation, or generating harmful content at scale. These advances underscore the critical importance of developing robust detection methods, implementing responsible access controls, and establishing clear ethical guidelines for deployment.

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

# A. Appendix

The supplementary material includes the following additional information:

- Sec. B provides the detailed hyper-parameters for the final BAR-FSQ and BAR models.

- Sec. C provides additional experimental results including BAR's results on the ImageNet-512 benchmark.

- Sec. D provides more visualization samples of BAR models.

# B. Hyper-parameters for Final BAR Models

*Table 5.* **Architecture configurations of BAR.** We follow prior works scaling up ViT (Dosovitskiy et al., 2021; Zhai et al., 2022) for different configurations.

| model | depth | width | mlp | heads | #params |
|-------|-------|-------|------|-------|---------|
| BAR-B | 24 | 768 | 3072 | 12 | 415M |
| BAR-L | 32 | 1280 | 5120 | 16 | 1108M |

*Table 6.* **Detailed hyper-parameters for final BAR-FSQ models.**

| config | value |
|--------|-------|
| *training hyper-params* | |
| optimizer | AdamW |
| learning rate | 1e-4 |
| weight decay | 0.0 |
| optimizer momentum | (0.9, 0.999) |
| total batch size | 256 |
| learning rate schedule | cosine decay |
| ending learning rate | 1e-5 |
| total epochs | 40 |
| warmup epochs | 2 |
| precision | bfloat16 |
| max grad norm | 1.0 |
| perceptual loss weight | 1.0 |
| clip l2 loss weight | 0.7 |
| gram loss weight | 10.0 |
| discriminator loss weight | 0.05 |
| discriminator kicks in epoch | 20 |
| reconstruction loss weight | 1.0 |
| training GPUs | 8 H200 |
| training time | 13 hrs |

The BAR model configuration is detailed in Tab. 5.

We list the detailed training hyper-parameters and sampling hyper-parameters for all BAR-FSQ, BAR models in Tab. 6 and Tab. 7, resptively.

# C. More Experimental Results

We provide additional experimental results on ImageNet-512 in Tab. 8, where BAR demonstrates clear advantages over other methods.

*Table 7.* **Detailed hyper-parameters for final BAR models.**

| config | value |
|--------|-------|
| *training hyper-params* | |
| optimizer | AdamW |
| learning rate | 4e-4 |
| weight decay | 0.03 |
| optimizer momentum | (0.9, 0.96) |
| total batch size | 2048 |
| learning rate schedule | cosine decay |
| ending learning rate | 1e-5 |
| total epochs | 400 |
| warmup epochs | 100 |
| annealing start epoch | 200 |
| annealing end epoch | 300 |
| precision | bfloat16 |
| max grad norm | 1.0 |
| dropout rate | 0.0 (B) / 0.2 (L) |
| attn dropout rate | 0.0 (B) / 0.2 (L) |
| class label dropout rate | 0.1 |
| training GPUs | 16 H200 (B) / 32 H200 (L) |
| training time | 20 hrs (B) / 32 hrs (L) |
| *sampling hyper-params w/ CFG* | |
| guidance schedule | linear |
| temperature | 2.5 (B) / 3.0 (L) |
| guidance scale | 5.0 (B) / 5.3 (L) |
| bits unmasking schedule | [2,2,5,7] |
| *sampling hyper-params w/o CFG* | |
| temperature | 2.0 (B) / 2.4 (L) |
| bits unmasking schedule | [4,4,4,4] |

*Table 8.* **ImageNet-1K** $512 \times 512$ **generation results.** We report metrics with classifier-free guidance. BAR only adopts a simple linear guidance schedule, with no need for auto-guidance (Karras et al., 2024) from an external model that is used by other state-of-the-art methods (Zheng et al., 2025b). Due to computational constraints, we only train the model for 200 epochs.

| method | #params | FID↓ | IS↑ |
|--------|---------|------|-----|
| VQGAN (Esser et al., 2021) | 227M | 26.52 | 66.8 |
| MaskGiT (Chang et al., 2022) | 227M | 7.32 | 156.0 |
| DiT (Peebles & Xie, 2023) | 675M | 3.04 | 240.8 |
| DiMR (Liu et al., 2024) | 525M | 2.89 | 289.8 |
| VAR (Tian et al., 2024) | 2.3B | 2.63 | 303.2 |
| REPA (Yu et al., 2025b) | 675M | 2.08 | 274.6 |
| xAR (Ren et al., 2025) | 608M | 1.70 | 281.5 |
| RAR (Yu et al., 2025a) | 1.5B | 1.66 | 295.7 |
| DDT (Wang et al., 2025a) | 675M | 1.28 | 305.1 |
| RAE (Zheng et al., 2025b) | 839M | 1.13 | 259.6 |
| BAR-L (ours) | 1.1B | **1.09** | **311.1** |

# D. Visualization on Generated Samples

We provide visualization results in Fig. 7, Fig. 8, Fig. 9, Fig. 10, Fig. 11, Fig. 12, Fig. 13, Fig. 14, Fig. 15, Fig. 16, Fig. 17, and Fig. 18.

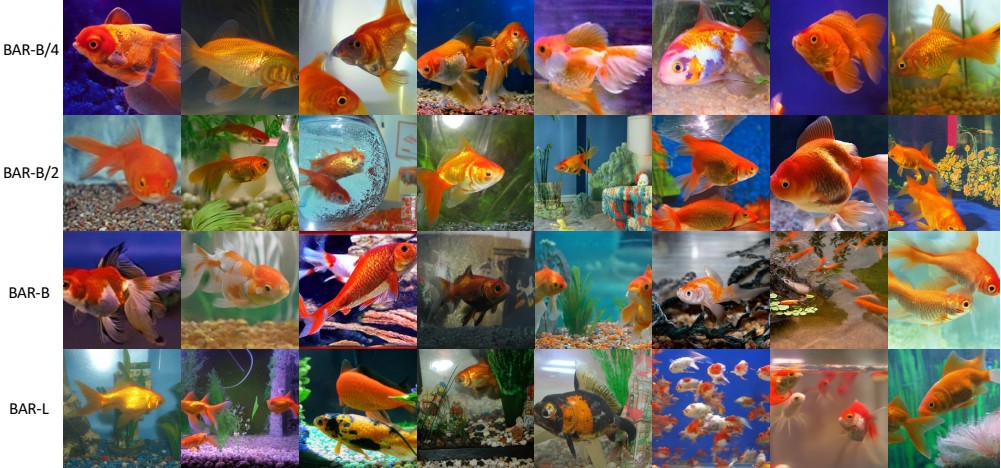

*Figure 7.* **Visualization samples from BAR models.** BAR is capable of generating high-fidelity image samples with great diversity. class idx 1: "goldfish, Carassius auratus".

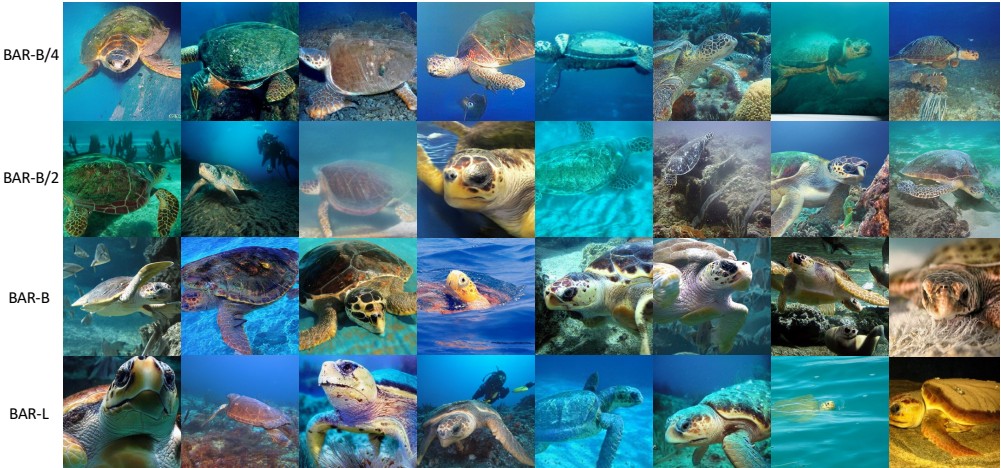

*Figure 8.* **Visualization samples from BAR models.** BAR is capable of generating high-fidelity image samples with great diversity. class idx 33: "loggerhead, loggerhead turtle, Caretta caretta".

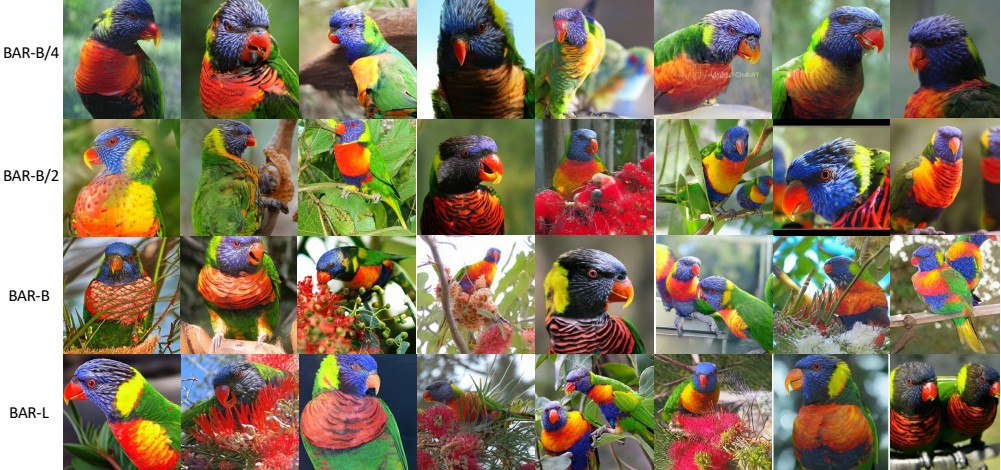

*Figure 9.* **Visualization samples from BAR models.** BAR is capable of generating high-fidelity image samples with great diversity. class idx 90: "lorikeet".

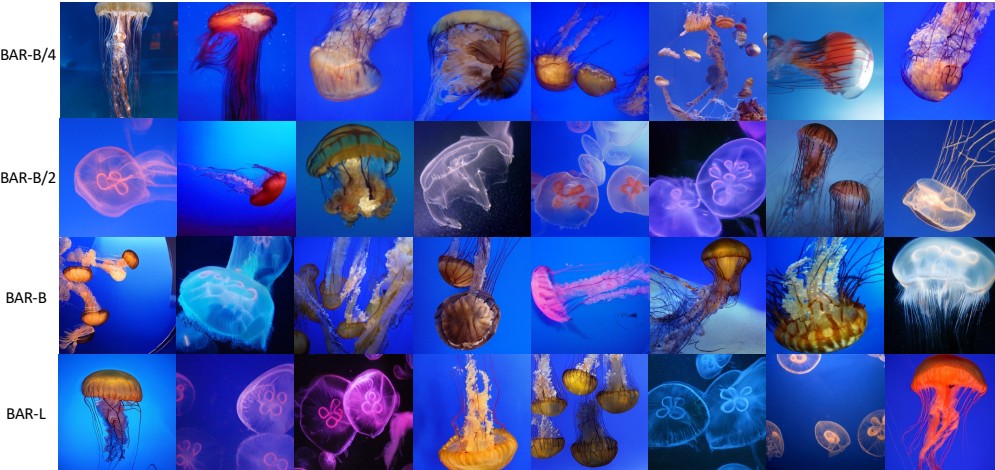

*Figure 10.* **Visualization samples from BAR models.** BAR is capable of generating high-fidelity image samples with great diversity. class idx 107: "jellyfish".

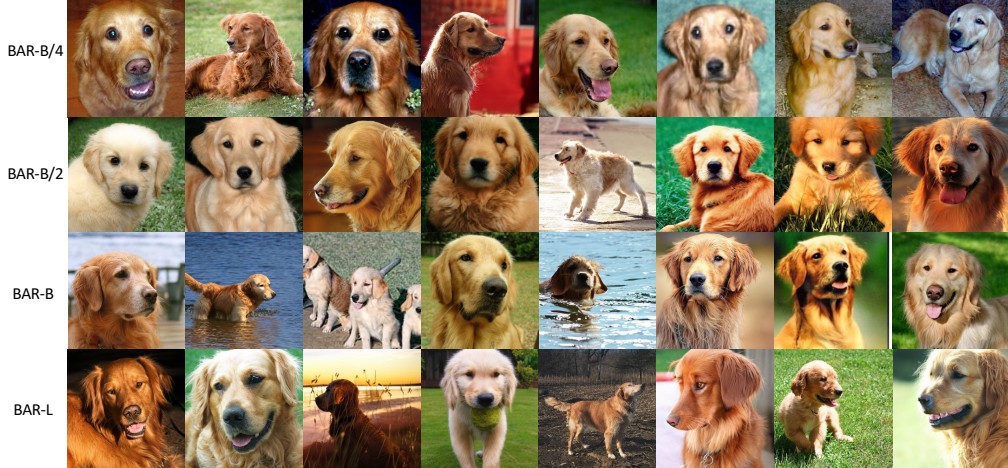

*Figure 11.* **Visualization samples from BAR models.** BAR is capable of generating high-fidelity image samples with great diversity. class idx 207: "golden retriever".

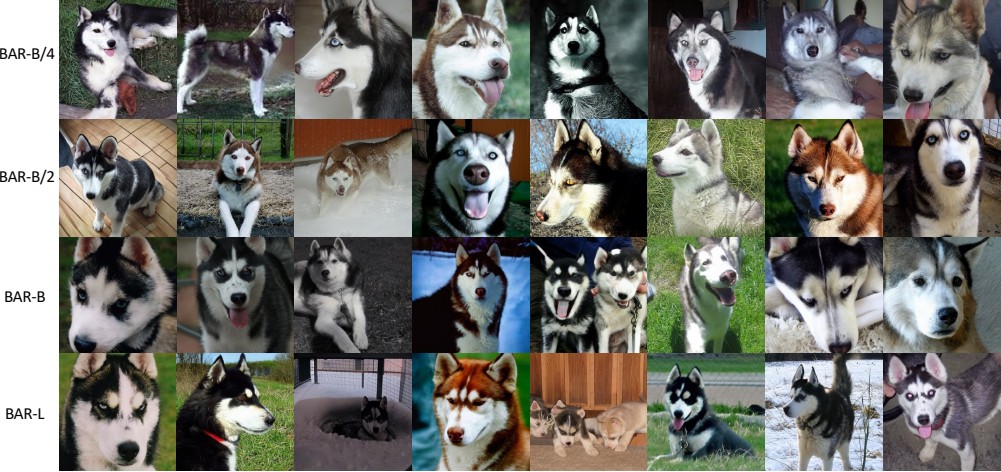

*Figure 12.* **Visualization samples from BAR models.** BAR is capable of generating high-fidelity image samples with great diversity. class idx 250: "Siberian husky".

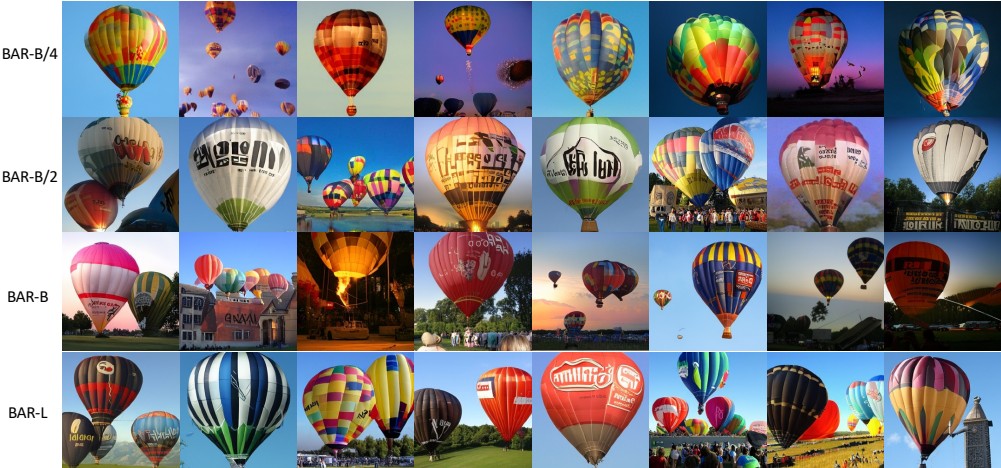

*Figure 13.* **Visualization samples from BAR models.** BAR is capable of generating high-fidelity image samples with great diversity. class idx 417: "balloon".

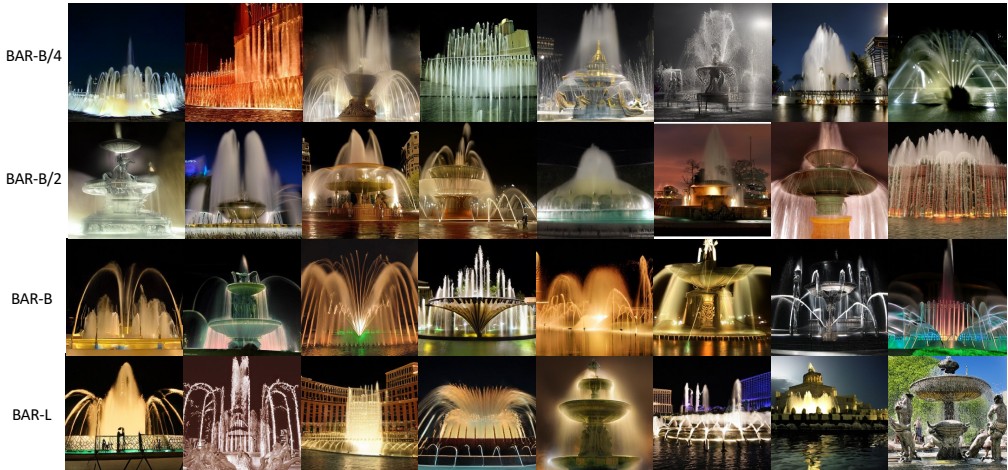

*Figure 14.* **Visualization samples from BAR models.** BAR is capable of generating high-fidelity image samples with great diversity. class idx 562: "fountain".

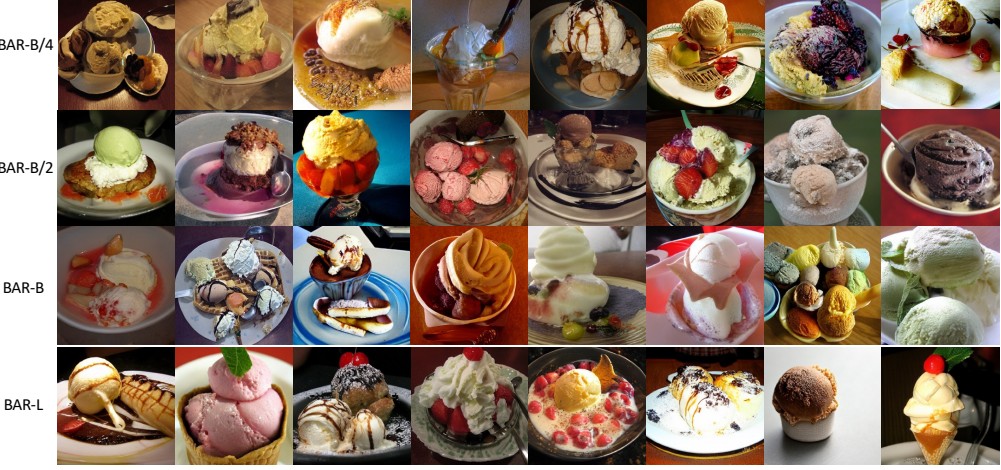

*Figure 15.* **Visualization samples from BAR models.** BAR is capable of generating high-fidelity image samples with great diversity. class idx 928: "ice cream, icecream".

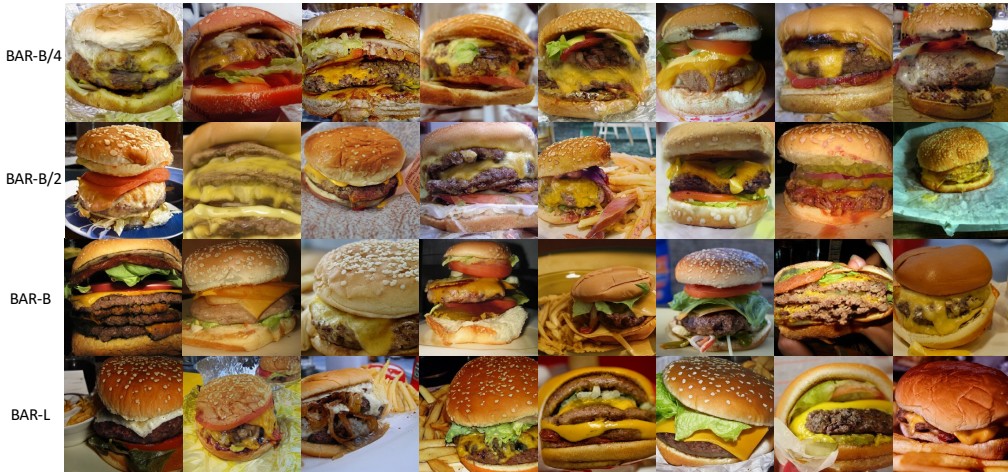

*Figure 16.* **Visualization samples from BAR models.** BAR is capable of generating high-fidelity image samples with great diversity. class idx 933: "cheeseburger".

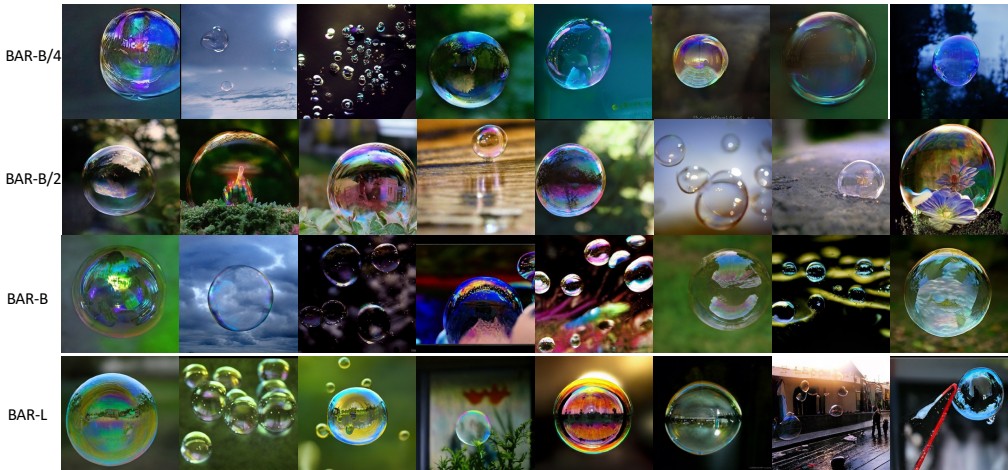

*Figure 17.* **Visualization samples from BAR models.** BAR is capable of generating high-fidelity image samples with great diversity. class idx 971: "bubble".

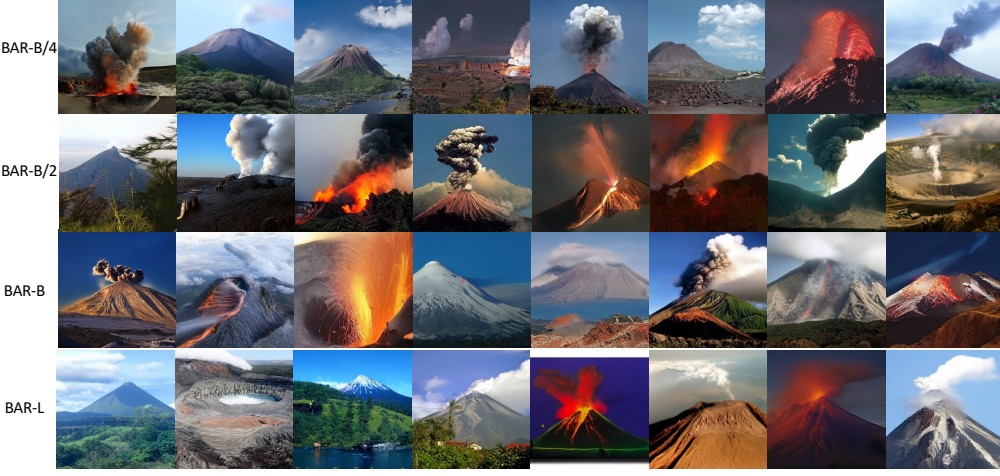

*Figure 18.* **Visualization samples from BAR models.** BAR is capable of generating high-fidelity image samples with great diversity. class idx 980: "volcano".

