# OpenReview forum: "Autoregressive Image Generation with Masked Bit Modeling"
_ICML.cc/2026/Conference — ICML 2026 regular_

### Official Review · Reviewer_mG3w · 2026-02-16

**Soundness:** 2
**Presentation:** 3
**Significance:** 3
**Originality:** 2
**Overall Recommendation:** 4
**Confidence:** 5

**Summary:**

The paper introduces BAR, a scalable image generation framework designed to support arbitrary codebook sizes. The authors investigate the performance gap between discrete and continuous tokenizers, analyzing why discrete variants often lack expressivity. To address this, they propose a framework that integrates autoregressive and masked modeling to generate tokenized images. Experimental results demonstrate that BAR achieves a state-of-the-art gFID of 0.99 on the ImageNet-256 dataset, while efficiency comparison confirm the model's superior sampling speed.

**Compliance With Llm Reviewing Policy:**

Affirmed.

**Key Questions For Authors:**

- Could the authors provide a performance comparison under an aligned hyperparameter search space with the baselines? Specifically, to decouple the benefits of the proposed sampling modification from potential over-tuning, could the authors evaluate BAR and the baselines using a fixed sampling configuration (e.g., linear scheduling, temperature=1.0, and a range of CFG scales from 0.0 to 10.0)? Without a standardized comparison, it is difficult to see whether the reported improvements stem from the proposed MBM or from setup-specific hyperparameter optimization.
- Could the authors provide a clearer guideline for selecting sampling hyperparameters across different contexts? Given the sensitivity observed in Table 7, are the current hyperparameter sets robust when: (a) applied to different checkpoints of the same model size? (b) transferring the framework to a new dataset?
- Regarding Figure 6, what is the underlying cause for the performance divergence as vocabulary size increases? The data shows that increasing the vocabulary size improves rFID but degrades gFID for the MBM head. Could the authors provide further analytical or experimental explanation for this trend?
- How does the MBM head compare to the decomposed head approach commonly used in LFQ (lookup-free quantization) literature? In many large-vocabulary contexts, people handle high-dimensional spaces by decomposing a single large head into multiple smaller heads (e.g., representing $2^{18}$ as two $2^9$ heads, potentially with autoregressive dependencies). Have the authors compare the MBM head against this standard decomposition strategy? what are the advantages of MBM over a standard multi-head decomposition in terms of efficiency and generation quality?

**Limitations:**

yes

**Strengths And Weaknesses:**

**Strengths**
- The comparative analysis between continuous and discrete tokenizers provides a well-justified foundation for the proposed method.
- The introduction of MBM heads balance the trade-off between the memory overhead of linear heads and the performance limitations of bit heads.
- The BAR framework demonstrates superior results, achieving improved image synthesis quality (FID) and inference throughput (images/sec).

**Weaknesses**
- The core components, i.e., bit-wise quantization and autoregressive generation, have been explored in prior literature. The discussion can be improved from a more explicit analysis of its theoretical properties or a clearer technical distinction from existing bit-based generative models.
- The method appears to require distinct hyperparameter configurations for different setups. As shown in Table 7, the CFG scale, temperature, and unmasking schedules vary based on model size and the application of CFG. This reliance on fine-tuned settings suggests that the sampling process may require extensive manual intervention, potentially limiting its practical application.
- Evaluation is restricted to the ImageNet dataset. Since the primary contributions are largely engineering-focused, the lack of testing on diverse datasets (e.g., COCO or specialized domains) leaves the robustness of the framework unverified.

---

> ### Author Rebuttal · Authors · 2026-03-31
>
> We thank the reviewer for the thoughtful questions and suggestions. We address each point below.
>
> **W1: Core components explored in prior literature.**
>
> We agree that FSQ and autoregressive generation are not new individually, and we do not claim them as contributions. Our main contributions are: (1) rethinking the discrete-continuous gap through the lens of bit budget, showing that discrete tokenizers are not intrinsically inferior but rather under-allocated in bits; (2) proposing MBM head as a scalable prediction head that makes generation with large discrete codebooks practical. Please also see our response to Reviewer M3iq (W2-P1) for a detailed comparison to MaskBit, Infinity, and MAR. We will strengthen the technical distinction from prior works in the revised paper.
>
>
> **W2, Q1, Q2: Hyperparameter sensitivity across setups.**
>
> The CFG scale and temperature are obtained via hyperparameter search with Ray Tune for all configurations. To directly address the concern, we provide the gFID landscape for BAR-B across a grid of CFG and temperature values:
>
> | temp \ cfg | 4.0 | 4.5 | 5.0 | 5.5 | 6.0 | 7.0 | 8.0 | 9.0 | 10.0 |
> | :---- | :---- | :---- | :---- | :---- | :---- | :---- | :---- | :---- | :---- |
> | 2.25 | 1.34 | 1.35 | 1.33 | 1.33 | 1.30 | 1.31 | 1.35 | 1.34 | 1.35 |
> | 2.50 | 1.16 | 1.16 | **1.13** | 1.14 | 1.17 | 1.20 | 1.24 | 1.29 | 1.36 |
> | 2.75 | 1.60 | 1.52 | 1.50 | 1.48 | 1.45 | 1.46 | 1.51 | 1.60 | 1.64 |
>
> The landscape is smooth and robust. Within the range of temp=[2.25, 2.5] and cfg=[4.0, 10.0], all configurations achieve gFID≤1.36. The optimal setting (temp=2.5, cfg=5.0, gFID=1.13) is not a narrow spike but part of a broad basin.
>
> We also note that optimal sampling hyperparameters vary significantly across methods: DiT uses cfg=1.5, VAR uses cfg=2.0+top-k=600, MAR uses cfg=3.2+temp=1.0, MAGVIT2 uses cfg=25+temp=15, MaskBit uses cfg=5.8+temp=10.3, RAR uses cfg=8.0+temp=1.02. There is no universal set of hyperparameters, and fixing identical values across all methods would not be a fair comparison. We perform extensive hyperparameter search with Ray Tune **for all methods** in our ablation studies to ensure fair comparison. That said, exploring universal hyperparameters is an interesting future direction.
>
>
> **W3: ImageNet-only evaluation.**
>
> ImageNet-256 is the standard benchmark used by all prior works in our comparison (VAR, MAR, VA-VAE, DDT, RAE, etc.), and we follow this convention. Additional ImageNet-512 results are provided in Table 8 of the Appendix, where BAR achieves a new state-of-the-art performance. We appreciate the suggestion to explore broader domains. Tasks like T2I primarily require dataset curation and large compute, which we see as promising future work.
>
>
> **Q3: rFID-gFID divergence in Figure 6.**
>
> Please see our response to Reviewer awRm (W1) for a detailed discussion. In short: the reconstruction-generation trade-off is universal across tokenizer paradigms; MBM head shifts the sweet spot significantly rightward (gFID degrades gracefully: 1.19→1.20→1.37→1.67 from 2^16 to 2^64); and stronger semantic regularization at higher bits is a promising unexplored direction.
>
>
> **Q4: MBM vs decomposed/multi-head approach (LFQ-style).**
>
> Great question. LFQ's approach of splitting the vocabulary into sub-groups sits between the bits head and linear head. The key limitation is that groups are predicted independently, losing inter-group bit dependencies. MBM captures these through progressive unmasking, where each step conditions on previously unmasked bits, similar to how autoregressive factorization outperforms independent prediction in language modeling. Table 1 supports this: independent bit prediction (bit head) yields inferior gFID vs MBM at all codebook sizes. For the rebuttal, we trained this decomposed head baseline (splitting 2^16 into two independent 2^8 groups), achieving **gFID=2.40, IS=273.9**. Compared to Table 1 at 16 bits: the decomposed head (gFID 2.40) outperforms the bit head (gFID 3.93) but is significantly worse than MBM (gFID 1.19), confirming that MBM's progressive unmasking provides a clear advantage over independent group prediction. We will include this ablation study in the revision.

---

> > ### Author Rebuttal · Reviewer_mG3w · 2026-04-02
> >
> > I thank the authors for their thorough response, which has adequately addressed my concerns. I will maintain my positive assessment of this paper.

---

### Official Review · Reviewer_awRm · 2026-03-02

**Soundness:** 4
**Presentation:** 4
**Significance:** 4
**Originality:** 4
**Overall Recommendation:** 5
**Confidence:** 4

**Summary:**

This paper presents an effective solution for empowering discrete generative models to surpass continuous counterparts.The authors pointed out that rather than the discussion between continuous or discrete generation, the number of bits for the latent space is the key influencer of tokenizer performance. To validate this observation, the authors made an attempt to scale the codebook size with an exceptionally large FSQ tokenizer. To adapt generation systems with such a large codebook, the authors proposed a novel discrete masked bit modeling head, enabling a new sota of less than 1 FID on ImageNet 256. Additional to performance, the authors also proposed parallel decoding designs to significantly speed the proposed model to > 400 imgs/s.

**Compliance With Llm Reviewing Policy:**

Affirmed.

**Final Justification:**

In this paper, the authors presents an effective solution for empowering discrete generative models to surpass continuous counterparts through scaling mask bit modeling. Overall, the motivation are clear and experiments solid. I believe that this paper should be accepted.

**Key Questions For Authors:**

See weaknesses.

**Limitations:**

yes

**Strengths And Weaknesses:**

Strengths:
1. Clear motivation and observation: the authors insightfully identified that the root cause to suboptimal in discrete generation systems.
2. Solid experiments and ablations: the authors conducted extensive analysis and ablation studies and presented clear and solid results on the effect of scaling and individual designs. The main result of <1.0 gFID is also very strong.
3. Impressive efficiency: Beyond strong performance, the authors also enabled efficient decoding via token-shuffling, enabling throughput of up to >400 imgs/s while still maintaining competitive performance.

Weaknessess:
1. The generation performance curve in FIg.6 seems to indicate that despite performance improvements at codebook size < 2^16, scaling beyond this point creates degradations to performance. Does this indicate potential limitations of the proposed generator/tokenizer, or is it an inherent conflict of the two-stage modeling paradigm? It would be best if more detailed analysis is conducted.

---

> ### Author Rebuttal · Authors · 2026-03-31
>
> We thank the reviewer for the positive assessment and insightful question.
>
> **W1: Performance degradation beyond codebook size 2^16 in Figure 6.**
>
> There is always a sweet spot between reconstruction and generation quality, as discussed in prior literature (LFQ, VA-VAE), regardless of paradigm. Better reconstruction does not always translate to better generation, as the latent space becomes harder to model. What matters is where this sweet spot lies: MBM head shifts it significantly rightward compared to linear and bit heads (Figure 6). Concretely, MBM head's gFID with CFG evolves as 1.19 (2^16) → 1.20 (2^18) → 1.37 (2^32) → 1.67 (2^64), showing graceful degradation in regimes where linear heads go OOM and bit heads suffer severely (e.g., bit head gFID=3.93 at 2^16). Even at 2^18, gFID=1.20 remains firmly in SOTA range. The "degradation" is relative to BAR's own peak, not to competing methods.
>
> We also note that all Table 1 experiments use the same 415M BAR-B generator. Scaling to BAR-L (1.1B) achieves gFID=0.99, suggesting that larger generators can push the sweet spot further. Additionally, existing literature shows that stronger semantic regularization (e.g., representation alignment in VA-VAE, REPA) helps make high-capacity latent spaces more generation-friendly. We did not explore such techniques at larger bit counts, and we believe this is an interesting future direction orthogonal to our contributions.
>
> The paper's core claims remain well-supported: (a) discrete tokenizers are not intrinsically inferior, the gap stems from an insufficient bit budget (Fig. 4); (b) MBM achieves a better reconstruction-generation trade-off than alternative prediction heads (Fig. 6). We appreciate the suggestions and will expand this discussion in the revised paper.

---

> > ### Author Rebuttal · Reviewer_awRm · 2026-04-01
> >
> > Thanks to the authors for the thorough rebuttal and my concerns have been fully addressed. I will be maintaining my score.

---

### Official Review · Reviewer_5TpV · 2026-03-07

**Soundness:** 3
**Presentation:** 3
**Significance:** 2
**Originality:** 2
**Overall Recommendation:** 5
**Confidence:** 5

**Summary:**

This paper investigates the performance gap between discrete and continuous visual tokenization pipelines. The authors argue that the gap is primarily due to differences in compression ratio (measured via a "Bit Budget" metric), not an intrinsic limitation of discrete representations. They show that scaling codebook size closes this gap. To handle the resulting vocabulary scaling problem in generation, they propose BAR (masked Bit AutoRegressive modeling), which replaces the standard linear prediction head with a Masked Bit Modeling (MBM) head that progressively unmasks bits of each token. BAR achieves a state-of-the-art gFID of 0.99 on ImageNet-256.

**Compliance With Llm Reviewing Policy:**

Affirmed.

**Final Justification:**

Authors provided detailed ablation answering the questions and concerns. This research demonstrates interesting paths in scaling discrete generation models and achieving superior generation quality compared to their continuous counterparts at lower budget.

**Key Questions For Authors:**

1) Can authors present results of direct comparison to continuous VAEs, trained on imagenet? For instance can they train SD VAE on imagenet for a few epochs so the comparison becomes fair?

2) How much of BAR-FSQ's reconstruction quality comes from the pretrained SigLIP2 encoder versus codebook scaling? Can you provide an ablation with a randomly initialized encoder at multiple codebook sizes?

3) Can authors clarify further how the CFG is done in this work? is guidance applied at the transformer logits, at the bit-level predictions during each unmasking step, or both? Results seem to be very much dependent on the CFG and this deserves more clarification

**Limitations:**

The limitation is not discussed in the work. In conclusion authors can describe whether they faced any challenges by going from index prediction to the bit prediction.

**Strengths And Weaknesses:**

Strengths
s1) Clear and well-motivated framing. The Bit Budget analysis (Section 3.2–3.3) is a useful contribution. Unifying discrete and continuous tokenizers under a common information-theoretic measure is simple but effective, and the systematic scaling experiments in Figure 4 are convincing.

s2) Strong empirical results. A gFID of 0.99 on ImageNet-256 is a great result for a discrete model, and the throughput comparisons in Table 4 are impressive. BAR-B at 415M parameters outperforms RAR at 1.5B parameters, demonstrating strong parameter efficiency. The efficient variants (BAR-B/2, BAR-B/4) offer a compelling quality-speed Pareto frontier.

s3) Thorough ablations. The ablation studies in Table 2 systematically examine masking strategy, head size, sampling steps, and token-shuffling. The paper covers a reasonable design space and the conclusions are well-supported.


Weaknesses
w1) The Bit Budget comparison is not rigorous. Eq. 4 assigns 16 bits per channel to continuous tokenizers based on floating-point precision, but effective entropy after KL regularization is far lower. This systematically overestimates continuous tokenizers' information usage, making discrete tokenizers appear artificially efficient. Without effective entropy estimates, the central claim that discrete tokenizers "beat" continuous ones at equal bit budgets is not established.

w2) No controlled tokenizer comparison. All continuous VAE baselines (SD-VAE, MAR-VAE, VA-VAE) are taken from prior work with different architectures, training recipes, and critically, different training data. SD-VAE was trained on a broad web dataset, not ImageNet it is easy to empirically see that finetuning SD-VAE on ImageNet dramatically improves rFID and LPIPS. Meanwhile, BAR-FSQ uses a pretrained SigLIP2 encoder, CLIP alignment, DINO discriminator, and modern loss combinations, all trained on ImageNet. Without training a continuous VAE under identical conditions (same backbone, same data, same losses), the claimed superiority of discrete tokenizers is confounded by engineering and data advantages.

w3) The overal novelty of the work is incremental. The MBM head applies masked generative modeling at the bit level, a natural extension of MaskGIT/MaskBit. The differences from MaskBit (predicting bits vs. indices) and Infinity (removing the external bit-corrector) are modest. The "paradigm shift" framing oversells what is a clean but incremental engineering contribution.

w4) There is no clear indication of faster convergence of this method. This claims goes unsupported from the presented results with no training curves or wall-clock comparisons to substantiate this.

---

> ### Author Rebuttal · Authors · 2026-03-31
>
> We thank the reviewer for the detailed and constructive analysis.
>
> **W1: Bit Budget comparison not rigorous (effective entropy).**
>
> We agree that discrete and continuous tokenizers follow different distribution priors, and KL regularization makes the effective entropy discussion nuanced. The bit budget serves as a practical proxy metric. It provides a more fair basis for comparison than the conventional practice of comparing across fundamentally different compression ratios.
>
> That said, in practice KL regularization is trained with very small weight (e.g., 1e-6 in SD-VAE/VA-VAE), which has marginal impact on the latent distribution. Moreover, even if per-channel entropy is somewhat reduced by KL, the spatial extent of the latent space (e.g., 256 tokens x 16 channels) is still fully utilized, meaning the total information capacity remains substantial. We will add a more detailed discussion of this point in the revised paper.
>
>
> **W2, Q1, Q2: No controlled tokenizer comparison.**
>
> We kindly note that both MAR-VAE and VA-VAE in our comparisons are already trained on ImageNet. The "advanced training recipe" components are all from existing literature widely used across both paradigms: semantic encoder initialization (VA-VAE, RAE, etc.), DINO discriminator (VAR, RAE, etc.). These are not unique to our method.
>
> We further ran new ablations (BAR-FSQ-16bits) isolating the effect of each component (see response to Reviewer M3iq W1 for details):
>
> | Setting | rFID |
> | :---- | :---- |
> | so400m, pretrained, CLIP loss (reported) | 1.02 |
> | so400m, pretrained, no CLIP loss | 0.85 |
> | so400m, scratch, CLIP loss | 0.90 |
> | so400m, scratch, no CLIP loss | 1.12 |
> | base-86M, pretrained, CLIP loss | 1.17 |
> | base-86M, pretrained, no CLIP loss | 0.94 |
> | base-86M, scratch, CLIP loss | 0.91 |
> | base-86M, scratch, no CLIP loss | 1.11 |
>
> Even without pretrained weights or CLIP loss, or smaller model size, BAR-FSQ remains competitive. We also trained a continuous variant of our tokenizer (same encoder, decoder, losses, and hyper-params, only replacing FSQ with a continuous bottleneck + KL regularization) to directly isolate the quantization strategy:
>
> | Quantizer | bit budget | rFID |
> | :---- | :---- | :---- |
> | FSQ (discrete, 64-bit) | 16384 | 0.50 |
> | FSQ (discrete, 128-bit) | 32768 | 0.38 |
> | FSQ (discrete, 256-bit) | 65536 | 0.33 |
> | VAE (continuous, KL=1e-6) | 65536 | 0.45 |
>
> Under identical training conditions, the discrete tokenizer matches the continuous variant with fewer bits (FSQ 64-bit rFID=0.50 vs VAE rFID=0.45), and clearly outperforms it at higher budgets (FSQ 128-bit rFID=0.38, FSQ 256-bit rFID=0.33), further confirming our conclusion from the paper. We will include this comparison in the revision.
>
>
> **W3: Incremental novelty (MBM as extension of MaskGIT/MaskBit).**
>
> While MBM shares the masking concept with MaskGIT, the contribution lies in how it is applied: (1) MaskGIT/MaskBit operate at the *token* level (predicting indices), while MBM operates at the *bit* level within each token, a fundamentally different decomposition enabling vocabulary-agnostic scaling. (2) MBM is formulated as a *conditional generation* task (with timestep conditioning, iterative refinement) rather than classification, consistently outperforming bit heads that also predict bits but without the generative framework (Table 1). (3) This makes MBM the first prediction head that scales smoothly to 2^64 and beyond, which neither MaskGIT nor MaskBit can achieve.
>
>
> **W4: Convergence claim.** We demonstrate this with both 80-epoch and 400-epoch results in Table 3, following the convention of prior works (VA-VAE, REPA, DDT, RAE). BAR achieves better performance at both training durations, with and without guidance, supporting the claim of faster convergence.
>
>
> **Q3: CFG clarification.** The CFG scale decays across the transformer's AR rollout steps and remains constant for the corresponding MBM head at each token position. The transformer only outputs latent conditions for MBM unmasking, so CFG is applied on the MBM head side. We will open-source all the codes (including training and inference), allowing the community to fully reproduce the results.
>
>
> **Limitations.** BAR presents a promising direction for unified multi-modal modeling under discrete tokens, but we did not explore this due to time and resource constraints. Improving the reconstruction-generation trade-off within the BAR framework is also worth studying. We believe these are promising future directions that remain underexplored. We will add a dedicated limitations discussion in the revised paper.

---

> > ### Author Rebuttal · Reviewer_5TpV · 2026-04-02
> >
> > I appreciate the authors for detailed ablations. This have answered the concerns and I am happy to raise my score.

---

### Official Review · Reviewer_M3iq · 2026-03-15

**Soundness:** 4
**Presentation:** 3
**Significance:** 4
**Originality:** 3
**Overall Recommendation:** 5
**Confidence:** 5

**Summary:**

This paper studies the size of discrete latent space in autoregressive models and its effect on reconstruction and generation quality. From the view on the latent space size, esp. with bit metric, the author performs throughout comparison with diffusion models and AR models, together with their underlying continuous VAEs and discrete tokenizers. The authors propose to use extended but binary latent discrete space in discrete tokenizers to maximize the latent bits and train a discrete tokenizer with that. Therefore, the authors propose a masked bit modeling head as a bit generator to predict these codes for autoregressive generation models. The resulted AR generation models excel at ImageNet generation benchmark, especially with smaller parameter counts.

**Compliance With Llm Reviewing Policy:**

Affirmed.

**Final Justification:**

Authors has answered my concerns in details and thus I remain positive to this paper.

**Key Questions For Authors:**

See weaknesses.

**Limitations:**

Yes. The limitations and potential negative societal impact of their work is included.

**Strengths And Weaknesses:**

Strengths:
1. The most strong and appealing part of this paper, in my opinion, is the generation result of BAR models. To my best knowledge, BAR for the first time outperforms (conventional) diffusion models as a discrete and AR generation models on ImageNet benchmark, reaching a gFID 0.99. When with less parameters, 415M, BAR-B also acheives a record-breaking gFID for discrete generation models. This particularly indicates the presented BAR method is a strong method.

1. The authors also perform extensive large-scale ablation studies to show what limits the generation capability with discrete latent space. These experiment designs should, of course, be encouraged for a paper. Broad audience can learn a lot of details from these experimental results and one can reprodure results from experiment details as described in this paper.

Weaknesses:
1. I think the most vulnerable part of this paper might be the study on the discrete tokenizer, especially in Figure 4. According to the experiment section, the authors use a SigLIP-400M as a backbone as the initialized encoder of BAR-FSQ tokenizer. This, however, serves a strong feature/resprentation space and might be a factor to well scale up with the discrete latent space, especially with per-bit binary assumptions. However, rivals in Figure 4, do not generally enjoy the strong representation encoder as initialization and this could serve as an unfair comparison. The authors are encouraged to train a small BAR-FSQ variants totally from scratch to show the trendency of reconstruction abilities.

1. The second vulnerable part, from my view, is the close similarity to MaskBit and Infinity. Through stated and clarified in related work, the connection between BAR and them is still too close, I would rather say. As my best knowledge, MaskBit performs MaskGIT but on binary views of tokens (MaskGIT on tokens * bits, not as stated by authors "predicting index") while BAR performs autoregressive modeling on token level but also perform MaskGIT on bits in individual tokens (AR on tokens, MaskGIT on bits). Inifinity performs VAR logic on tokens and linear head on bits. While strong results achieved, the authors are strongly encouraged to state the difference and connection between BAR and others. AND, the statement claimed on linear head on Figure 5 seems too strong without full evidence (why linear head enjoys good performance GIVEN it does not model bit-bit corelations? why bit head enjoys good scalability GIVEN training might be harder? This also applied to MBM head's scalability as the training with large bit counts remains unexplored). Also, BAR (AR on tokens, MaskGIT on bits) somehow resembles MAR (partially-AR on tokens, diffusion on token embeddings/bits).

1. Small concern: the latency of MBM head since it requires multiple forwards as MaskGIT logic on every token generated. Could the author also analyse this part of latency since this is inevitable on every token in AR procedure as token-wise sampling.

---

> ### Author Rebuttal · Authors · 2026-03-31
>
> We thank the reviewer for the positive assessment and thoughtful feedback.
>
> **W1: SigLIP-400M initialization as an unfair advantage.**
>
> We appreciate this concern and have run a comprehensive ablation: encoder size (SigLIP2-so400m vs SigLIP2-base-86M) x initialization (pretrained vs scratch) x CLIP loss (with vs without). Results (rFID, EMA, 200K steps, 16-bit tokenizer):
>
> | Setting | rFID |
> | :---- | :---- |
> | so400m, pretrained, CLIP loss (reported) | 1.02 |
> | so400m, pretrained, no CLIP loss | 0.85 |
> | so400m, scratch, CLIP loss | 0.90 |
> | so400m, scratch, no CLIP loss | 1.12 |
> | base-86M, pretrained, CLIP loss | 1.17 |
> | base-86M, pretrained, no CLIP loss | 0.94 |
> | base-86M, scratch, CLIP loss | 0.91 |
> | base-86M, scratch, no CLIP loss | 1.11 |
>
> Model size, pretrained weights, and CLIP loss are not crucial for reconstruction, as shown in the table. The role of semantic encoders is primarily to regularize the latent space rather than to boost reconstruction, as widely adopted in prior works across both paradigms (REPA, VA-VAE). Since these factors do not substantially impact tokenizer quality, our core conclusion remains intact: discrete tokenizer reconstruction keeps improving with more bits and can match or surpass continuous counterparts given sufficient bit budgets.
>
> We will include the full ablation in the revised paper.
>
>
> **W2-P1: Differences from MaskBit/Infinity/MAR.** We first note that our paper's novelty extends beyond Masked Bit Modeling (MBM). The core contribution is rethinking the discrete-continuous gap through bit budget analysis, and MBM is then proposed as the scalable solution this insight demands. Regarding specific methods: MaskBit adopts LFQ and feeds bit tokens to the masked image modeling generator at input level, but still relies on a linear head predicting codebook indices at inference (Fig. 5 (b) in main paper), which does not solve vocabulary scaling, while BAR employs an autoregressive modeling generator that handles bit tokens at both input and output level. Infinity uses a bits-head (Fig. 5 (c)) and also employs an external bit-corrector for post-processing, while BAR provides a simple and unified solution with the MBM head. Regarding the resemblance to MAR: MAR uses diffusion loss over continuous token embeddings, while BAR operates in discrete bit space with progressive unmasking; the two are fundamentally different generation mechanisms. We acknowledge the importance of and inspiration from these prior works. As shown in Table 1, both linear heads (MaskBit) and bits-heads (Infinity) are suboptimal at scale, and in Table 4 we show that BAR achieves better generation quality while being much faster than MAR, thanks to its discrete nature. We will include more detailed comparisons in the revised paper.
>
>
> **W2-P2: Linear head being strong.** Our linear head variant is built on the SOTA discrete baseline RAR, so our linear head results essentially reproduce RAR but with improved architecture (RoPE, SwiGLU, RMSNorm) that has been widely adopted in recent diffusion models (e.g., LightningDiT, DDT, RAE, JiT). The strong linear head results actually show that MBM's improvements come on top of an already very strong baseline, making the gains meaningful rather than inflated by a weak starting point.
>
>
> **W2-P3: Scalability.** Apologies for any confusion. By scalability we mean both *feasibility* (can the head handle large vocabularies?) and *performance* (does generation quality hold?). While both bits head and MBM are feasible at extreme scales, the bits head degrades significantly as vocabulary grows (Table 1: gFID>10 without CFG at 10 bits). MBM maintains strong performance. We experiment up to 64 bits (~1.84x10^19 codebook size, Table 1), while still achieving SOTA-level generation quality. We will clarify these concepts in the revision.
>
>
> **W3: MBM head latency.** The extra cost is marginal since the multiple forward passes only occur within the MBM head (a lightweight MLP), not the full autoregressive transformer. We benchmark BAR-B throughput across different MBM steps:
>
> | MBM Steps | 1 | 2 | 3 | 4 (default) |
> | :---- | :---- | :---- | :---- | :---- |
> | img/s | 26.31 | 25.23 | 24.54 | 24.33 |
>
> Going from 1 step to 4 steps (our default setting) only reduces throughput by ~8%, while providing substantial quality gains (Table 2c). BAR-B at 4 steps (24.33 img/s) is 3.68x faster than RAE at matching gFID (Table 4).

---

> > ### Author Rebuttal · Reviewer_M3iq · 2026-04-02
> >
> > Thanks to the author for the detailed response and insightful discussions, which also addressed my concerns. I will maintain my positive score on this paper.

---

### Decision · Program_Chairs · 2026-04-30

**Decision:**

Accept (regular)

**Comment:**

The paper studies how discrete latent space size affects image generation. The authors propose using a larger binary latent space and introduce BAR, which combines autoregressive and masked bit modeling. Their method achieves impressive results on ImageNet with better quality and faster generation using fewer parameters.